# Glaucoma through Animal’s Eyes: Insights from the Evolution of Intraocular Pressure in Mammals and Birds

**DOI:** 10.3390/ani12162027

**Published:** 2022-08-10

**Authors:** Watcharapong Hongjamrassilp, Roger Zhang, B. Natterson-Horowitz, Daniel T. Blumstein

**Affiliations:** 1Department of Marine Science, Faculty of Science, Chulalongkorn University, Phayathai Road, Bangkok 10330, Thailand; 2Department of Ecology and Evolutionary Biology, University of California Los Angeles, 621 Young Drive South, Los Angeles, CA 90095, USA; 3David Geffen School of Medicine at UCLA, Division of Cardiology, 650 Charles E. Young Drive South, A2-237, Los Angeles, CA 90095, USA

**Keywords:** IOP, evolutionary medicine, comparative medicine, glaucoma, life history

## Abstract

**Simple Summary:**

Understanding how a disease evolved across the animal kingdom could help us better understand the disease and might lead to novel methods for treatment. Here, we studied the evolution of glaucoma, an irreversible eye disease, in mammals and birds, by studying the evolution of intraocular pressure (IOP), a central driver of glaucoma, and searching for associations between life history traits and IOP. Our results revealed that IOP is a taxa-specific trait that is higher in some species than in others. Higher IOPs appear to have evolved multiple times in mammals and birds. Higher IOPs were found in mammals with higher body mass and in aquatic birds. We also found that higher IOPs evolved through stabilizing selection, with the optimum IOP in mammals and birds being 17.67 and 14.31 mmHg, respectively. This supports the hypothesis that higher IOPs may be an adaptive trait for certain animals. Focusing on species with higher IOPs but no evidence of glaucoma may help identify glaucoma-resistant adaptations, which could be developed into human therapies.

**Abstract:**

Glaucoma, an eye disorder caused by elevated intraocular pressure (IOP), is the leading cause of irreversible blindness in humans. Understanding how IOP levels have evolved across animal species could shed light on the nature of human vulnerability to glaucoma. Here, we studied the evolution of IOP in mammals and birds and explored its life history correlates. We conducted a systematic review, to create a dataset of species-specific IOP levels and reconstructed the ancestral states of IOP using three models of evolution (Brownian, Early burst, and Ornstein–Uhlenbeck (OU)) to understand the evolution of glaucoma. Furthermore, we tested the association between life history traits (e.g., body mass, blood pressure, diet, longevity, and habitat) and IOP using phylogenetic generalized least squares (PGLS). IOP in mammals and birds evolved under the OU model, suggesting stabilizing selection toward an optimal value. Larger mammals had higher IOPs and aquatic birds had higher IOPs; no other measured life history traits, the type of tonometer used, or whether the animal was sedated when measuring IOP explained the significant variation in IOP in this dataset. Elevated IOP, which could result from physiological and anatomical processes, evolved multiple times in mammals and birds. However, we do not understand how species with high IOP avoid glaucoma. While we found very few associations between life history traits and IOP, we suggest that more detailed studies may help identify mechanisms by which IOP is decoupled from glaucoma. Importantly, species with higher IOPs (cetaceans, pinnipeds, and rhinoceros) could be good model systems for studying glaucoma-resistant adaptations.

## 1. Introduction

Glaucoma, an eye disease associated with elevated intraocular pressure (IOP), is a leading cause of irreversible visual loss in humans. Elevated IOP is a central driver of the degeneration of optic nerves that results in gradual visual impairment. To date, no treatment can completely reverse the damage to optic nerves, making glaucoma the second leading cause of blindness and the first leading cause of irreversible blindness in humans after cataracts [1,2]. While it is a common visual pathology in our species, glaucoma is not uniquely human. Many animals, especially domestic pets (e.g., cats and dogs) and livestock (e.g., cattle), have been reported to have glaucoma [3,4]. However, because of the limitations of technology and the feasibility of diagnosing glaucoma in different species of animals, the prevalence of this disease in other species of animals, especially wildlife, is still undocumented [5]. Understanding how glaucoma evolved across the animal kingdom might shed light on susceptibility and adaptation to the disease [6,7].

To investigate the presence of glaucoma in animals, we must first find ways to identify the pathology in nonhuman animals. In humans, diagnosing glaucoma requires measuring IOP, conducting a visual field test, and conducting an optical coherence tomography (OCT) test [8]. However, conducting a similar set of assessments among a wide range of nonhuman animals in varied settings is logistically impossible [5]. For captive animals, veterinarians use a tonometer during a screening test to measure IOP [9]. This helps determine the risk of having glaucoma. Nevertheless, because of limited data on reports of glaucoma in non-humans, we used IOP as a proxy to assess the chance of having glaucoma in non-human animals.

IOP in humans is associated with many factors, such as blood pressure, age, diet, and genetics [10,11,12]. These traits vary across the animal kingdom. For example, giraffes (*Giraffa camelopardalis*) have a systolic blood pressure that can exceed 280 mmHg. On the other end of the spectrum, the range of normal systolic blood pressures in modern reptiles can range from 40–60 mmHg. Human systolic blood pressures, by way of contrast, are 120 mmHg [13,14]. The average lifespan of bowhead whales (*Balaena mysticetus*) is 211 years, while humans’ average lifespan is 72 years, and small rodents, such as Chinese hamsters (*Cricetulus griseus*), live 2–3 years [15,16]. The variation in these traits and their correlations with IOP have rarely been investigated in non-humans. A comparative study of the association between these traits and IOP will allow us to better understand how animals may have adapted to survive under relatively high IOP levels.

Research suggests that extreme recreational activities such as SCUBA diving and bungee jumping increases IOP and should be avoided in patients with glaucoma [17]. However, many non-humans engage in similarly intense activities as part of their daily lives. For instance, pinnipeds may engage in extremely deep foraging dives. Some pinniped species, such as elephant seals (*Mirounga angustirostris*), can dive more than 1500 m, which is almost 50 times deeper than humans can dive without specialized equipment [18]. Such diving is similar to SCUBA diving in humans, and a recent study suggests that some species of pinnipeds have glaucoma [19]. Some terrestrial birds, such as Rüppell’s vultures (*Gyps rueppellii*), fly up to 11,300 m and must deal with substantially different air pressures on such flights [20]. Similarly, to bungee jumping, how can birds deal with this rapid change in air pressure, and does the abrupt change in air pressure affect their IOP? These questions could be addressed if we first understand the pattern and evolution of IOP in animals.

We conducted a comparative phylogenetic study, to understand the evolution of IOP in mammals and birds. First, we mapped IOP traits on mammal and bird phylogenies and conducted ancestral state reconstruction to understand the evolutionary pattern of IOP. Second, we studied the correlation between life history traits and IOP. Due of the tight linkage between IOP and glaucoma in humans, the results of this exploratory research could improve our understanding of the susceptibility to, and protection from, glaucoma in animals.

## 2. Materials and Methods

### 2.1. Comparative IOP Dataset

We used the results of a previous systematic review of IOP and supplemented it with new searches. The initial IOP dataset in birds and mammals [21] contained IOP data found from searches of PubMed, Scopus, and BioOne databases, from their inception to 31 August 2015. We used the same search terms, and inclusion and exclusion criteria from the original data base and searched the Web of Science (all databases) from August 2015 to February 2020.

### 2.2. Phylogenetic Tree Preparation

We obtained phylogenetic trees of mammals and birds from http://vertlife.org/ (accessed on 27 February 2021). The VertLife database has two types of phylogenetic trees, a “complete” tree and a “sequenced species only” tree. Both types of trees provide different evolutionary relationships and tree topologies [22,23]. Therefore, we combined both types of trees to generate a maximum consensus tree. For mammalian trees, we combined 1000 mammal birth-death node-dated completed trees (5911 species) with 1000 mammal birth-death node-dated DNA-only trees (4098 species). For bird trees, we combined 1000 trees of “all species birds” with 1000 trees of “sequence species”. After concatenating both datasets in each mammal and bird tree, we generated a maximum clade credibility (MCC) tree using the function *maxCladeCred* in *phangorn* R package (version 2.7.0) (R Core Team, Vienna, Austria) [24]. These mammalian and avian trees were used throughout the following analysis.

### 2.3. Studying the Evolution of IOP

To understand the evolution of IOP, we mapped the IOP and reconstructed the ancestral state of IOP on the mammal and bird MCC tree using a maximum-likelihood approach. As IOP is a continuous trait, we fitted three models of continuous trait evolution, prior to conducting an ancestral state reconstruction (ASR). The three models included: (1) a Brownian model, a model that explain trait evolution by random drift or multiple selective pressures [25]; (2) an early-burst model, a model that infers adaptive radiation by rapid change in traits early in time, which then slow as time progresses [26], and (3) an Ornstein–Uhlenbeck model, a model that occurs when there is stabilizing selection around an optimal value [27,28]. We conducted model fitting using the function *fitContinuous* in *geiger* R package (version 2.0.7) and compared the Akaike information criterion (AIC) score of these three models. We used the lowest AIC score to select which of the three models best-fit the trait variation. Then, we simulated how IOP evolved with model parameters obtained from model fitting—the σ^2^ parameter for Brownian models and the α and θ parameters for OU models—for 100 generations in R, to visualize constraints on trait evolution. Finally, we mapped IOP data and implemented ancestral state reconstruction using the functions *fastBM* and *contMap* in the *phytools* R package (version 0.7-70) [29].

### 2.4. Studying Life History Correlates of IOP

We explored life history traits that have been reported to be associated with an increase in IOP. Many studies revealed that body mass is slightly associated with IOP in both non-humans and humans [21,30]. Therefore, we included average body mass (kg) in this study and hypothesized that body mass might be positively associated with IOP levels across mammals and birds.

There is some discussion about whether SCUBA diving is associated with increased IOP in humans [17,31,32]. Do diving animals also experience high IOP, as in SCUBA divers? This question has never been addressed. We, therefore, included the habitat (aquatic/terrestrial) that animals inhabit and the maximum diving depth (meters) for diving species in this study. We postulated that aquatic mammals and birds might have higher IOPs than those found in terrestrial species.

Blood pressure correlates with IOP in humans [33]. Patients with hypertension often have high IOP and a concomitant increased risk of glaucoma [34]. Moreover, hypertension is also associated with age, and older adults tend to have a higher risk of having hypertension and high IOP [35]. Hence, we included systolic blood pressure (mmHg), the pressure exerted when a heart beats, and maximum longevity (years) from captive animals as life history traits.

Some complementary and alternative medicine studies identified a link between diet and IOP levels in humans who mainly consume fruits and vegetables [12,36]. Given the variety of diets in non-humans, we included diet type (herbivore, carnivore, and omnivore) as another life history trait.

We obtained the life history trait information of the species for which we had IOP information from various databases. For average body mass (kg), habitat (aquatic, terrestrial), and diet type (herbivore, carnivore, omnivore), we used the information from https://animaldiversity.org (accessed on 1 March 2021). For maximum longevity (years), we obtained the information from the AnAge database (https://genomics.senescence.info/species/index.html) (accessed on 1 June 2022), which reports maximum captive longevity. For average blood pressure (mmHg) and maximum diving depth (meters), we obtained the information from literature searching through Google Scholar and Web of Science. Since there are few data of avian blood pressure, we omitted the analysis of bird’s blood pressure. In addition, the data from average body mass and maximum longevity were not normally distributed. We, therefore, transformed the data using log_10_ transform prior to analyzing the data.

As species share common ancestors, and as they might not have evolved independently [37], we then tested for a phylogenetic signal, a proxy of statistical dependence among species’ traits, due to their phylogenetic relationship [38]. To do so, we fitted each life history trait with four different modes of evolution: (1) Brownian motion, (2) Ornstein–Uhlenbeck (OU), (3) Pagel’s lambda, and (4) white noise (a non-phylogenetic model). We used the function *gls* in the *nlme* package in R and function *corBrownian*, *corMartins*, and *corPagal* as a correlation parameter for the first three models, respectively. For the white noise model, we used only the function *gls*. We then compared the AIC score of each model and interpreted the model with the lowest AIC.

The type of tonometer (applanation and rebound tonometer) and sedation used during IOP measurements influence IOP [21,39]. However, our IOP data were obtained with various types of tonometer, and some of the IOPs were measured while animals were sedated. To control the effect of these two factors in our study, we used phylogenetic generalized least squares (PGLS) to investigate the effect of tonometer and sedation on IOPs in our dataset. In mammals, we a conducted PGLS analysis by fitting the type of tonometer and sedation with IOPs. However, in birds, we only fitted type of tonometer with IOPs, because most IOP data from birds were obtained without sedation.

To investigate the relationship between life history traits and IOP, we fitted a PGLS model with log_10_(average body mass) because a preliminary analysis in mammals showed that IOP was correlated with log_10_(average body mass). Then we added each of the following variables to the log_10_(average body mass) only model: (1) habitat (aquatic, terrestrial), (2) diet type (herbivore, carnivore, omnivore), (3) average blood pressure, (4) log_10_(maximum longevity), and (5) maximum diving depth. For the bird group, log_10_(average body mass) did not correlate with IOP. Therefore, we fitted three PGLS models. The first model had three independent variables log_10_(average body mass), habitat, and diet type, and all data were available for all three traits. Since all data were not available for all species, we fitted two additional models using all the available data: (1) log_10_(maximum longevity), and (2) maximum diving depth.

## 3. Results

### 3.1. Evolution of IOP

We studied IOP evolution using 63 species (28 families, 10 orders) of mammals and 43 species (13 families, 11 orders) of birds, and found that IOP levels in mammals and birds evolved under the Ornstein–Uhlenbeck model (Table 1). This result means that, rather than drifting around randomly, IOP evolved towards an optimum value. The optimum value (θ) of IOP in mammals and birds based on OU model fitting were 17.67 and 14.31 mmHg, respectively. To visualize this, Figure 1B and Figure 2B and Figure 1C and Figure 2C show the simulation of IOP evolution under Brownian and OU models, respectively. When considering the simulation from OU model, trait values at present were clustered more compared to the simulation from Brownian model, indicating “stabilizing selection”.

In humans, an IOP over 22 mmHg defines ocular hypertension [40]. In mammals, the ancestor state reconstructions suggested that higher IOPs evolved at least six times (Figure 1A) independently. This can be seen two times in the order Artiodactyla (buffalos, and dolphins and porpoises), one time in the order Perissodactyla (rhinoceroses and horses), two times in the order Carnivora (pinnipeds, lions, and cheetahs), and once in the order Diprotodontia (koalas). While IOPs below 6.5 mmHg are considered to be pathologically low in humans, known as hypotony [41], we found that lower IOPs evolved at least two times independently in the order Rodentia (lowland pacas and hamsters).

In birds, the ancestral state reconstruction suggested that birds initially had a low IOP, and that higher IOPs independently evolved at least two times (Figure 2A), appearing one time in the order Sphenosciformes (penguins) and one time in the order Accipitridae (eagles).

### 3.2. Life History Correlates of IOP

Even though much research reports that type of tonometer and sedation influence IOP levels, our PGLS analysis showed that in our IOP dataset, neither the type of tonometer (*p* = 0.219), nor sedation (*p* = 0.129) explained the variation in IOP in mammals (Table 2). Similarly, variation in IOP in our avian dataset was not explained by type of tonometer (*p* = 0.891) (Table 3).

In addition, our results revealed that IOP levels increased in mammals in association with body mass (*p* < 0.01) (Table 2). However, this association did not exist in birds (*p* = 0.678) (Table 3). Even though the results from non-PGLS in mammals showed that habitat (*p* < 0.01) and diet type (*p* < 0.01) were associated with IOP, our best supported models were those that incorporated phylogenetic information (Appendix A). When considering the PGLS results, we found no association between the aforementioned traits (including average blood pressure) and IOP (Table 2).

In birds, only the first model did not require phylogenetic information, and in this model, we found an association between habitat and IOP (*p* = 0.01) in birds. Aquatic birds that dive, such as penguins, have higher IOPs than non-diving birds (Table 3). We also found no relationship between log_10_(average body mass) (*p* = 0.678) or diet type (*p* = 0.806 and *p* = 0.537) and IOP. The other two best-supported models were those with phylogenetic information. In these models, we found no relationship between IOP and log_10_(maximum longevity) (*p* = 0.708) or maximum diving depth (*p* = 0.743) (Table 3).

## 4. Discussion and Conclusions

Our macroevolutionary study revealed that IOP, a trait that results from a combination of anatomical and physiological processes, evolved under the OU model in both mammalian and avian clades. This indicates that IOP may be under stabilizing selection. Based on the available data, our results reveal that the optimal IOP values in mammals and birds are 17.67 and 14.31 mmHg, respectively. These optimal values fall within the range of healthy IOP values in humans [40] and some non-human species [42]. The interspecific variation in IOP levels is notable, because the IOP levels in some species might be sufficient to induce pathology in humans. This suggests that evolved adaptations may exist in some species that confer resistance to glaucoma.

Zouache et al. found that IOP levels increased with the evolution of terrestrial animals [21]. This may have been influenced by the transition from lens-based optics in aquatic animals to cornea-based optics in terrestrial animals. This leads to an additional question: What are the IOPs levels of marine mammals, whose terrestrial ancestors “returned” to the water over 30 million years ago? Even though we found no association between IOP and habitat in mammals, our comparative study revealed that most marine mammals have a higher IOP (25–32.8 mmHg) than their ancestors (Figure 1A). This is similar to seabirds, which have IOPs ranging from 20.36–28.18 mmHg (Figure 2A). Nevertheless, the IOP data we have were measured when the animals were on land. Research on cetaceans found that when whales dive, they can reduce their heart rate (bradycardia) to 4–8 beats per minute (bpm) and increase it to 40 bpm at the surface (tachycardia) [43]. Emperor penguins have a similar diving response, where their heart rate is reduced to 5–10 bpm at the bottom of the ocean, from 200–240 bpm at the sea surface [44]. Since blood pressure is associated with elevated IOP in humans, this might be one of the possible mechanisms in marine mammals and seabirds to reduce IOP while diving underwater with a high atmospheric pressure. However, more research, and specifically measurements of IOP conducted under water, is needed to evaluate this hypothesis.

Our research has a number of limitations. First, there are limited IOP data available for mammals and birds. We only have IOP data representing 3% of the total mammalian species and less than 1% of the total avian species. This small data set might affect statistical analysis and may lead to an incomplete interpretation. Moreover, IOP fluctuates within individuals; volume status, developmental phase, and other factors are among the influences of these shifts. Consequently, an analysis of mean IOP values cannot capture a full picture of IOP.

Second, because health status was not necessarily available, our inclusion criteria [21] for IOP values may have included some individuals with underlying ocular or other pathologies. However, our analyses are based on the assumptions that most of measured IOPs represented a “normal” state rather than a pathological one.

A third concern is the inconsistency of the tonometer that was used to measure the IOP. There are two main types of tonometry, rebound tonometry and applanation tonometry, and these methods may lead to different measurements [21]. This might increase the variance in the estimates IOP. Even though the type of tonometer did not explain variation in our IOP dataset, future studies should be aware of this issue when conducting comparative studies between multiple species.

Finally, corneal stiffness (e.g., central cornea thickness) has been shown to affect IOP levels when measuring IOP with a tonometer [45]. When measuring IOP with the same tonometer, thinner corneas give lower IOPs compared to thicker corneas [46]. Our study could not control for this factor because of the lack of corneal stiffness information in non-human animals. Thereby, it is essential to obtain corneal stiffness information while collecting IOP data in future studies, especially in wildlife. Since this was exploratory research, we suggest that more IOP data in other non-humans are needed to better understand the evolution of IOP and glaucoma in vertebrates.

The results from our study provide information regarding IOP as a species-specific characteristic and an understanding of the evolutionary pattern of IOP in mammals and birds. Future studies are needed to investigate how varying selective pressures shaped the morphological and physiological mechanisms that underlie IOP. In addition, the Krogh principle states that “for a large number of problems, there will be some animal of choice, or a few such animals, on which it can be most conveniently studied” [47]. We suggest that identifying species with higher IOPs could be a source of insights for developing novel approaches to glaucoma treatment and prevention. How, for example, do animals with IOPs that fall outside of the optimum value range, including cetaceans and rhinoceros, remain glaucoma-free? Awareness of species with normal visual function, despite higher IOPs, may spark interest in understanding the mechanisms that appear to confer this resistance. Moreover, since our research provides information regarding the macroevolution of IOP, future research could include a comparative genomic study of genes that might lead to elevated IOP or cause glaucoma across vertebrates.

## Figures and Tables

**Figure 1 animals-12-02027-f001:**
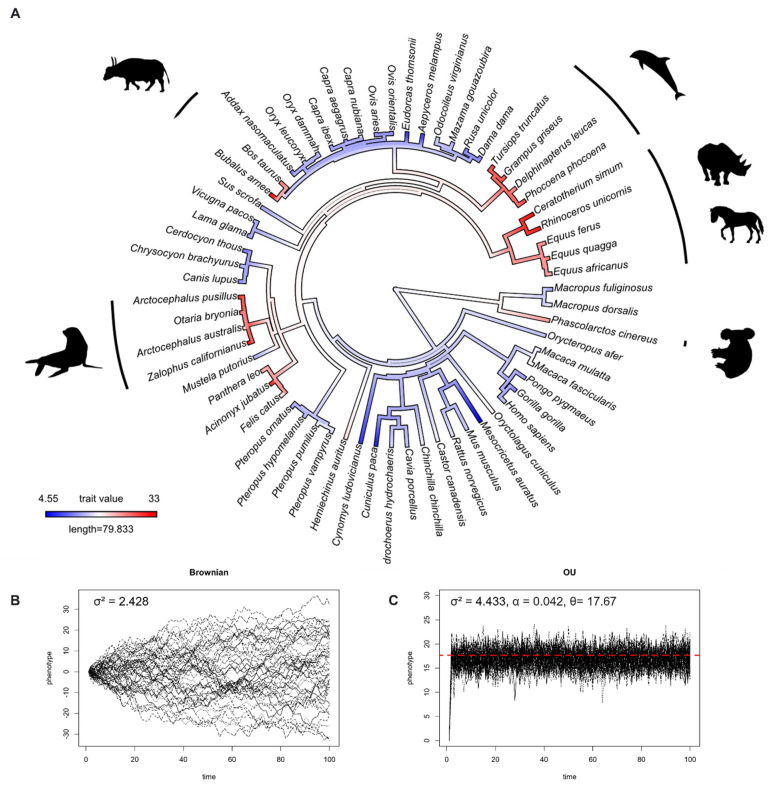
(**A**) Ancestral state reconstruction of intraocular pressure (IOP) from 63 species of mammals. Dark red color represents high IOP, and dark blue color represents low IOP. (**B**) Simulations of IOP evolution under Brownian model with evolutionary rate (drift) parameter (σ^2^). (**C**) Simulations of IOP evolution under the OU model with evolutionary rate (drift) parameter (σ^2^), pull rubber band (or pull towards optimum) parameter (α), and optimum trait value (θ). Red dashed line represents optimum value.

**Figure 2 animals-12-02027-f002:**
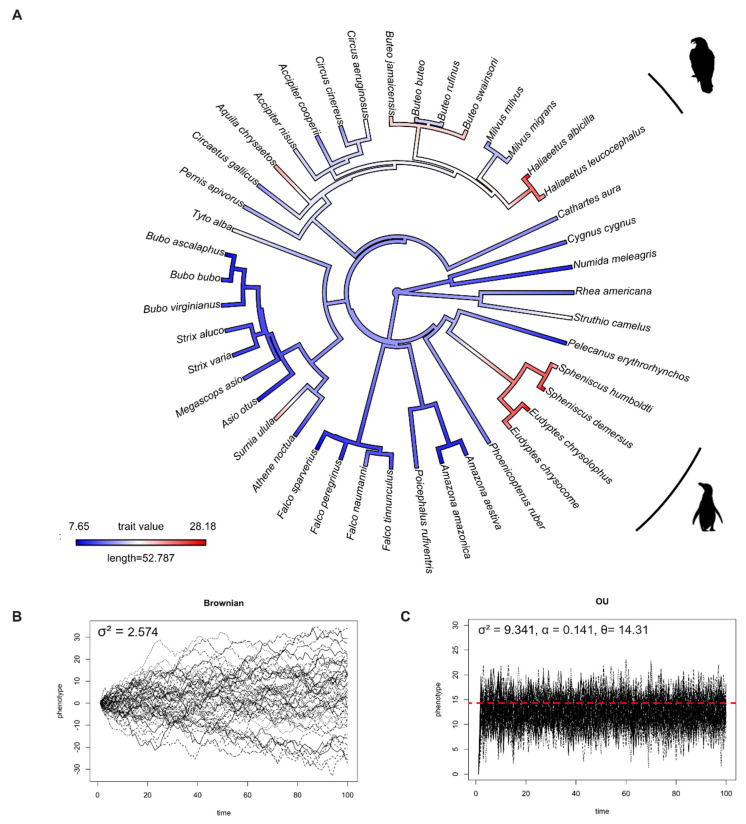
(**A**) Ancestral state reconstruction of intraocular pressure (IOP) from 43 species of birds. Red color represents high IOP, and blue color represents low IOP. (**B**) Simulations of IOP evolution under Brownian model with evolutionary rate (drift) parameter (σ^2^). (**C**) Simulations of IOP evolution under OU model with evolutionary rate (drift) parameter (σ^2^), pull rubber band (or pull towards optimum) parameter (α), and optimum trait value (θ). Red dashed line represents optimum value.

**Table 1 animals-12-02027-t001:** Comparison of phylogenetically signals of intraocular pressure (IOP) in mammals and birds using three modes of evolution (Brownian, Pagel’s lambda, Early-burst, and Ornstein-Uhlenbeck). Asterisks are the model with the lowest AIC value.

Model	Test Statistic	AIC	AICc
**Mammals**			
Brownian	K = 0.1856 *p* = 0.001 Sigma squared = 2.4279 Theta = 17.85	426.713	426.914
Early-Burst	Alpha = <0.001 Sigma squared = 2.4282 Theta = 17.58	428.714	429.121
Ornstein–Uhlenbeck	Alpha = 0.0425 Sigma squared = 4.4332 Theta = 17.67	407.164 *	407.571 *
**Birds**			
Brownian	K = 0.1554 *p* = 0.243 Sigma squared = 2.5747 Theta = 13.22	312.054	312.354
Early-Burst	Alpha = <0.001 Sigma squared = 2.5750 Theta = 13.22	314.050	314.670
Ornstein–Uhlenbeck	Alpha = 0.1408 Sigma squared = 9.3411 Theta = 14.31	276.187 *	276.802 *

**Table 2 animals-12-02027-t002:** Results of generalized least squares, with and without phylogenetic models, for life history traits in mammals. Asterisks represent significant results. For the habitat variable, terrestrial is the reference level. For the diet variable, carnivore is the reference level.

Variables	N	Non-Phylogenetic GLS Model		Phylogenetic Models			
		Estimate	*p*-Value	Estimate	*p*-Value	Model	α (O-U)/λ (Pagel)
**log_10_(average body mass)**	63	1.223	0.000 *	1.229	0.001 *	O-U	0.061
Type of tonometer	45	−2.019	0.433	−3.765	0.219	O-U	0.052
Sedation	50	3.880	0.076	2.844	0.129	O-U	0.041
**PGLS + log_10_(average body mass)**							
Habitat (Aquatic)	63	−7.771	0.001 *	−5.921	0.074	O-U	0.071
Diet (Herbivore)	63	−6.584	0.001 *	−3.885	0.139	O-U	0.071
Diet (Omnivore)	63	−4.879	0.052 *	−1.918	0.499	O-U	
Average blood pressure (systolic)	30	0.009	0.857	0.013	0.784	Pagel	0.498
log_10_(Maximum longevity)	48	1.053	0.614	−0.077	0.972	O-U	0.066
Maximum diving depth	9	0.002	0.792	0.002	0.792	O-U	3.975

**Table 3 animals-12-02027-t003:** Results of generalized least squares, with and without phylogenetic models, for life history traits in birds. Asterisks represent significant results. For the habitat variable, terrestrial is the reference level. For the diet variable, carnivore is the reference level.

Variables	N	Non-Phylogenetic GLS Model		Phylogenetic Models			
		Estimate	*p*-Value	Estimate	*p*-Value	Model	α (O-U)/λ (Pagel)
Type of tonometer	34	−2.340	0.529	0.347	0.891	Pegal	0.788
log_10_(average body mass) + habitat + diet	43					Pagel	0.705
Intercept		14.195	0.000 *	13.219	0.000 *		
log_10_(average body mass)		0.141	0.668	0.115	0.678		
Habitat (Aquatic)		10.207	0.000 *	11.777	0.011 *		
Diet (Herbivore)		−2.231	0.461	−0.959	0.806		
Diet (Omnivore)		−3.580	0.240	−2.270	0.537		
log_10_(Maximum longevity)	40	−0.383	0.857	0.640	0.708	Pagel	0.705
Maximum diving depth	4	−0.020	0.743	−0.020	0.743	O-U	3.958

## Data Availability

All data are provided in the Appendix A.

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
