# Peer review of "Glaucoma through Animal’s Eyes: Insights from the Evolution of Intraocular Pressure in Mammals and Birds"

_animals, 2022, doi:10.3390/ani12162027_

Round 1
Reviewer 1 Report
The authors present and interesting approach to documenting the evolutionary evidence for the risk/resistance to glaucoma development across the animal kingdom, using ‘normal’ intraocular pressure as a proxy. I commend them for their efforts. Their cumulative effort through analysis is a current gap in our field, and is needed. For the most part, the methodological and data portion of the study is very beneficial and well done, and is worthy of publication.
However, there are several fundamental problems with the manuscript. First, a critical error is not controlling for the method of tonometry. It is well recognized, both in the published literature and among the comparative ophthalmology community, that different tonometers in some species produce dramatically different results and cannot be scaled alongside different tonometers and different species. Clinically, they cannot even be compared in a single individual to properly determine normal vs. a glaucomatous state. In other species, the difference is much less. This will have to be addressed, and potentially controlled for, to make the data accurate and significant. Once incorporated, this dataset could be very impactful. Second, improving the discussion surrounding the data would greatly strengthen the manuscript, as well as benefit the potential readers. Some of the ideas should be elaborated on (e.g. underwater diving, etc.) whereas others should be given far less weight with a more comparative explanation to why it probably doesn’t matter (e.g. systemic blood pressure). I recommend the introduction and discussion be reformulated, restructured, and written in a way to highlight the truly interesting aspects of this dataset. Finally, there are frequent grammatical and conceptual errors, namely regarding glaucoma and other key ophthalmological topics.
My best recommendation is that the authors seek to include a veterinary ophthalmologist to help set-up, interpret, and provide discussion for this excellent dataset. It is all right here, the groundwork is laid - by working with someone with the particular skillset that a veterinary ophthalmologist has will likely bring it to full fruition.
Author Response
Reviewer 1
The authors present and interesting approach to documenting the evolutionary evidence for the risk/resistance to glaucoma development across the animal kingdom, using ‘normal’ intraocular pressure as a proxy. I commend them for their efforts. Their cumulative effort through analysis is a current gap in our field, and is needed. For the most part, the methodological and data portion of the study is very beneficial and well done, and is worthy of publication.
However, there are several fundamental problems with the manuscript.
First, a critical error is not controlling for the method of tonometry. It is well recognized, both in the published literature and among the comparative ophthalmology community, that different tonometers in some species produce dramatically different results and cannot be scaled alongside different tonometers and different species. Clinically, they cannot even be compared in a single individual to properly determine normal vs. a glaucomatous state. In other species, the difference is much less. This will have to be addressed, and potentially controlled for, to make the data accurate and significant. Once incorporated, this dataset could be very impactful.
Reply
[1] Thank you for pointing out this problem. We did an additional PGLS analysis to investigate the effect of type of tonometer (rebound VS applanation tonometer) on IOP in our dataset. Our results show that no significant variation in IOP was explained by tonometer type in these analyses. We now report these new results and comment on them in the discussion. Clearly, future studies must report tonometer type because we know that it does influence IOP measurements.
Second, improving the discussion surrounding the data would greatly strengthen the manuscript, as well as benefit the potential readers. Some of the ideas should be elaborated on (e.g. underwater diving, etc.) whereas others should be given far less weight with a more comparative explanation to why it probably doesn’t matter (e.g. systemic blood pressure).
Reply
We agree with your comments. We have searched for more information to further discuss our results. Unfortunately, little is known about the mechanism that controls the change of IOP in non-human animals. We view this paper as a first exploratory analysis and suggest that these results should stimulate other researchers to conduct more comparative analyses. In the future, we expect that such comparative studies will help us identify more species that can be studied in depth to better understand mechanisms that prevent animals from getting glaucoma.
I recommend the introduction and discussion be reformulated, restructured, and written in a way to highlight the truly interesting aspects of this dataset.
Reply
This manuscript aims to make readers aware of the use of macroevolutionary analyses in medical research (i.e., evolutionary medicine). We use glaucoma as a case study since no one has studied the evolution of glaucoma in this aspect before.
We intentionally wrote the introduction in a way to introduce the questions and tools used to ask macroevolutionary questions. Given the exploratory nature of this study and the lack of detailed information on glaucoma in non-humans, we were careful to not over-speculate. We hope that after these revisions, the reviewer finds the MS acceptable.
Finally, there are frequent grammatical and conceptual errors, namely regarding glaucoma and other key ophthalmological topics.
My best recommendation is that the authors seek to include a veterinary ophthalmologist to help set-up, interpret, and provide discussion for this excellent dataset. It is all right here, the groundwork is laid - by working with someone with the particular skillset that a veterinary ophthalmologist has will likely bring it to full fruition.
Reply
We have fixed grammatical errors when found and will certainly fix others if pointed out. We hope that this study has cast a light on the value of comparative studies to understand the evolution of glaucoma. We would be keen to collaborate with veterinary ophthalmologists in future studies.

Reviewer 2 Report
The manuscript presented by Hongjamrassilp et al. presents an interesting investigation into IOP distribution throughout the animal kingdom. The manuscript is well written and I only have a few comments:
- The mathematical modeling of evolutionary development is an interesting approach, but one wonders if more informative findings could have been found if the data would have been analyzed in smaller subgroups ( e.g. birds independent from mammals). After all, any number of factors might have driven speciation and IOP is likely to be a minor factor among these.
- A significant concern here is that the method(s) of IOP measurement is not reported and, given that the data was derived from numerous publications, likely varied among species. IOP is also affected by sedation which would have been used in some species, but not in others. Finally, most methods to measure IOP are influenced by corneal rigidity and measurements of eyes with stiff corneas will generate a falsely high IOP reading. This is noticeable among humans and could be a major source of error among species. After all, the eye is an optical system, distances and alignments matter, but this could also lead to a more rigid eye in e.g. diving species.
In summary, this is an interesting thought experiment that unfortunately does not lead to strong findings. This may be due, in part, to the difficulties to obtain the true IOP. This limitation should at least be throughly discussed.
Minor points:
- there appear to be occasional changes in font and/or size
-- first sentence Abstract: i believe that glaucoma is the second most common cause of irreversible blindness, not all blindness. Please check.
Author Response
Reviewer 2
The manuscript presented by Hongjamrassilp et al. presents an interesting investigation into IOP distribution throughout the animal kingdom. The manuscript is well written and I only have a few comments:
- The mathematical modeling of evolutionary development is an interesting approach, but one wonders if more informative findings could have been found if the data would have been analyzed in smaller subgroups (e.g. birds independent from mammals). After all, any number of factors might have driven speciation and IOP is likely to be a minor factor among these.
Reply
The MS separately analyzed birds’ and mammals’ IOP. We agree with you that when focusing on subgroups (e.g., only primates not all mammals), we might be able to find more informative results. Unfortunately, there are relatively few comparative IOP data (as we pointed out) which prevents us from further dividing up the dataset to conduct additional analyses. We hope that in the future this will be possible.
A significant concern here is that the method(s) of IOP measurement is not reported and, given that the data was derived from numerous publications, likely varied among species. IOP is also affected by sedation which would have been used in some species, but not in others. Finally, most methods to measure IOP are influenced by corneal rigidity and measurements of eyes with stiff corneas will generate a falsely high IOP reading. This is noticeable among humans and could be a major source of error among species. After all, the eye is an optical system, distances and alignments matter, but this could also lead to a more rigid eye in e.g. diving species.
Reply
Thank you for your useful comments. We conducted additional analyses to test the effect of sedation on our IOP dataset. We found no significant relationship between sedation and IOP level in our mammal IOP dataset. For birds, most of birds’ IOPs were investigated without sedation. Therefore, we assume that sedation does not affect birds’ IOP in our dataset.
Regarding corneal stiffness, we tried to obtain more information on cornea stiffness/rigidity/thickness and central corneal thickness. However, data on these features in non-human animals are scarce. Therefore, we could not control this factor in our study. We wrote about this issue in the discussion section. Future comparative studies would benefit from such data and we hope that this paper will stimulate additional studies.
In summary, this is an interesting thought experiment that unfortunately does not lead to strong findings. This may be due, in part, to the difficulties to obtain the true IOP. This limitation should at least be throughly discussed.
Minor points:
- there appear to be occasional changes in font and/or size
-- first sentence Abstract: i believe that glaucoma is the second most common cause of irreversible blindness, not all blindness. Please check.
Reply
Fixed all of them, thank you!

Reviewer 3 Report
Hongjamrassilp and co-authors, in “Glaucoma through animal’s eyes: Insights from the evolution of intraocular pressure in mammals and birds,” provide a compilation of interesting and valuable data regarding IOP in various species of animals. The analysis of associations is robust and useful.
There are some assumptions that may require further thought. Measuring IOP is a poor biomarker for glaucoma, with low sensitivity and specificity. It would be better if the authors abandoned the idea that they are assessing glaucoma and simply focus on the IOPs in different species and their associations with various physical and behavioral factors. However, altitude and air and water pressure do not affect IOP. SCUBA diving may increase IOP due to mask squeeze, but that is not what animals experience. Mountain climbers do not experience IOP change, as their entire body is exposed to the same atmospheric pressure.
The authors state that, “In humans, an IOP over 22 mmHg is considered to be elevated,” but that is not quite right. IOP > 22 is more than 2 SD above the mean, so considered high, not “elevated." By the same token, IOP <10 in humans is not hypotony, but is simply 2 SD below the mean IOP.
Overall this is an interesting and useful manuscript. The authors should be cautious in their interpretations of the meanings of their findings.
Author Response
Reviewer 3
Hongjamrassilp and co-authors, in “Glaucoma through animal’s eyes: Insights from the evolution of intraocular pressure in mammals and birds,” provide a compilation of interesting and valuable data regarding IOP in various species of animals. The analysis of associations is robust and useful.
There are some assumptions that may require further thought. Measuring IOP is a poor biomarker for glaucoma, with low sensitivity and specificity. It would be better if the authors abandoned the idea that they are assessing glaucoma and simply focus on the IOPs in different species and their associations with various physical and behavioral factors. However, altitude and air and water pressure do not affect IOP. SCUBA diving may increase IOP due to mask squeeze, but that is not what animals experience. Mountain climbers do not experience IOP change, as their entire body is exposed to the same atmospheric pressure.
Reply
Thank you very much for your critical comments.
First of all, we are aware that IOP alone is an inadequate proxy for glaucoma (precise diagnosis require visual field tests, or OCT). However, as stated in the introduction, identifying glaucoma with these methods is sometimes not feasible in non-human animals, especially wildlife. Therefore, we understand that at least IOP can be used as a proxy to estimate the chance of having glaucoma. Of course, this not 100% accurate, but as an initial exploratory analysis, we believe there is something to learn from developing a comparative understanding of IOP. Indeed, a primary aim of this manuscript is to raise awareness of this understudied research topic.
The authors state that, “In humans, an IOP over 22 mmHg is considered to be elevated,” but that is not quite right. IOP > 22 is more than 2 SD above the mean, so considered high, not “elevated." By the same token, IOP <10 in humans is not hypotony, but is simply 2 SD below the mean IOP.
Reply
We edited this part and added references regarding how high and low IOPs are identified.
Overall this is an interesting and useful manuscript. The authors should be cautious in their interpretations of the meanings of their findings.

Reviewer 4 Report
In the present research the authors investigated the evolution of the intraocular pressure in mammals and birds to understand the evolution of glaucoma. The manuscript was comprehensive, However the following points are to be considered:
1. This is a systematic review, the type of the article should be modified to review rather than article type.
2. The reference styles in the text and the reference list are different. the style should be standardized.
3. Some traits were considered in the current studies, what about other factors like age, and gender.
Author Response
Reviewer 4
In the present research the authors investigated the evolution of the intraocular pressure in mammals and birds to understand the evolution of glaucoma. The manuscript was comprehensive, However the following points are to be considered:
- This is a systematic review, the type of the article should be modified to review rather than article type.
- The reference styles in the text and the reference list are different. the style should be standardized.
- Some traits were considered in the current studies, what about other factors like age, and gender.
Reply
[1] We will leave this decision to the editor.
[2] We reformatted it.
[3] We include age in the parameter “Maximum lifespan” and we did not include sex in our study because of the limitation of IOP data reported in the literature (which often did not report the sex of the animal studied). We recognize that these are shortcomings but are unavoidable and hope that this MS will stimulate additional comparative research which collects (and reports) additional data.

Round 2
Reviewer 1 Report
Thank you for all of your edits, the manuscript is very strong. I recommend it for publication.
Author Response
The reviewer agrees with this revised version of the manuscript.
Reviewer 4 Report
The authors reply to the comments is satisfying
Author Response

(The authors gave the same response as above.)
